# A Disability-Inclusive Healthcare-to-Well-Being Translational Science Framework

**DOI:** 10.3390/ijerph21010018

**Published:** 2023-12-22

**Authors:** Robin G. Lanzi, Riddhi A. Modi, James Rimmer

**Affiliations:** 1School of Public Health—Health Behavior, The University of Alabama at Birmingham, 1665 University Blvd., Birmingham, AL 35294, USA; rmodi@uabmc.edu; 2School of Health Professions—Occupational Therapy, National Center on Health, Physical Activity and Disability (NCHPAD), The University of Alabama at Birmingham, 3810 Ridgeway Drive, Birmingham, AL 35209, USA; jrimmer@uab.edu

**Keywords:** people with disabilities, healthcare, well-being, well-being promotion, health promotion, wellness, translational science, implementation science, strategic plan

## Abstract

The recent World Health Organization report on disability noted that people with disabilities (PWD) have many unmet health and rehabilitation needs, face numerous barriers to accessing healthcare and specialized services, and have overall worse health than people without disability. In view of this urgency to better identify and address health inequities systematically, we convened an expert panel of 14 stakeholders to develop a strategic plan that addresses this issue. The panel identified two major obstacles to quality healthcare services for PWD: (1) lack of coordination between the various healthcare sectors and community well-being programs and (2) substantial challenges finding and accessing healthcare services that meet their specific needs. The expert stakeholder panel noted that well-being self-management programs (both online and in person) that are easily accessible to PWD are critically needed. PWD must transition from being cared for as patients to individuals who are able to self-manage and self-advocate for their own health and well-being plans and activities. The proposed strategic plan offers providers and local communities a framework to begin addressing accessible and appropriate healthcare-to-well-being services and programs for PWD in managing their health in partnership with their healthcare providers.

## 1. Introduction

The second global report on health equity for persons with disabilities (PWD) noted that PWD experience greater health inequities and die 20 years younger compared to the general population [1]. The World Health Organization (WHO) report also noted that PWD have many unmet health and rehabilitation needs [2], face numerous barriers to accessing healthcare and specialized services, and have overall worse health than people without disability [1]. 

Meeting the holistic needs of people with disabilities (PWD) is of paramount importance as countless PWD are underserved and widely impacted by health disparities, secondary health conditions, and inadequate healthcare [3,4,5,6,7,8]. In the U.S., approximately 61 million Americans or one in four Americans report having a disability [9] and the prevalence of PWD is projected to continue to increase as people are living longer. Adults with disabilities are more likely to be obese (41.6% vs. 29.6%), smoke (21.9% vs. 10.9%), have heart disease (9.6% vs. 3.4%) and diabetes (15.9% vs. 7.6%) than adults without a disability [9,10]. Further, one in four adults with disabilities between 18–44 years of age report not having a usual healthcare provider and not being treated for an unmet healthcare need because of the cost in the past year [9]. 

The public health community is often not equipped to adequately support implementing inclusive well-being programs that meet the needs of PWD [11,12]. This results in the majority of PWD being unable to transition from the healthcare sector to well-being programs in the community [13,14]. Evidence-based well-being programs delivered and/or made available by public health communities that are developed specifically for the needs of PWD are lacking [2,15,16,17,18]. Hence, the existing publicly available health promotion programs that are currently available for PWD are woefully inadequate compared to those living without a disability, resulting in health disparities for PWD [11,12]. In fact, recently the National Institutes of Health made a landmark decision to designate people with disabilities as a population with health disparities [19].

As such, there is also a lack of information on connecting healthcare to well-being to support health improvements for PWD [20,21,22,23]. 

To advance health equity for PWD, a coordinated, intentional collaboration of healthcare sectors, public health communities, persons living with a disability must occur through an inclusive implementation science approach that supports the needs of PWD at all levels of the socio-ecological model (SEM) [24]. In view of the urgency to better address health disparities and health equity in an intentional systematic manner for PWD [25], this call-to-action paper presents an overarching framework with a detailed strategic plan developed by an expert stakeholder panel of researchers and thought leaders. The panel was tasked with identifying barriers and making recommendations for establishing a *Disability-Inclusive Healthcare-to-Well-Being Translational Science Framework*.

## 2. Materials and Methods

Two expert panels were convened and in-depth stakeholder interviews were conducted by a consulting firm.

### 2.1. Expert Panels

Stakeholders were selected based on one of the following criteria: (a) professional with a disability and expertise in health, wellness, and/or rehabilitation; (b) authority in disability research; (c) authority in healthcare and/or public health with expertise in disability; (d) person living with a disability and interest in providing feedback related to health and well-being. Stakeholders were invited to participate via a formal email describing the purpose of the strategic plan, time commitment, and desired outcome (i.e., a formal plan). All who were invited agreed to participate.

Two separate expert panels of researchers and thought leaders in the fields of health administration, public health, and rehabilitation science were convened to identify barriers to well-being initiatives for PWD. The experts were split into one of two committees based on their area of expertise. One group focused on healthcare and the other group focused on public health. As previously noted, critically important and intentionally, the expert panels were comprised of researchers and thought leaders that included PWD. The expert panels were tasked with identifying the current challenges to community-based health promotion (i.e., well-being), the unique needs of the target population, and the existing barriers that prevent them from accessing these initiatives. Both committees developed a person-centered continuum of care that focused on increasing access to well-being promotion for this underserved group. Each committee had a specific emphasis. One committee focused on removing barriers to well-being initiatives at the health provider level through analysis of the existing health care delivery system and its limitations. The other committee focused on removing barriers to well-being initiatives at the community engagement level. Expert panel members are listed in the Acknowledgements section.

Both expert panels developed their own recommendations for bridging the gap in the continuum of care with the goal of PWD being able to self-manage or co-manage their own healthcare/post-rehabilitation well-being. Both sets of recommendation were created in collaboration with experts who have a disability and are currently working in the field. The concept of “nothing about us without us” was a central theme during the planning and creation of this strategic plan. 

### 2.2. In-Depth Stakeholder Interviews

When the need for the strategic plan was identified, one of the authors [JR] approached an independent consulting firm (Clarus Consulting, Birmingham, AL, USA) who provided a prospectus and cost estimate. Once approved, the consulting firm began their work.

The consulting firm conducted one-on-one interviews with nine key stakeholders and critical partners to inform the strategic planning process. A member of the consulting firm met with three experts in disability to develop a set of questions that each interviewee would be asked: (1) What are the unique needs of the target population as it relates to healthcare and well-being?; (2) What type of opportunities are needed to improve their overall care?; and (3) What are the challenges for implementing a healthcare-to-well-being program for PWD? Upon completion of the interviews, the consultant noted that the perspectives of PWD informed each phase of the plan’s creation and eventual implementation. A thematic analysis was implemented which began with coders reviewing the transcripts, developing an initial set of codes, then taking the codes and grouping them into themes, and finally determining a final coding and thematic structure through an iterative process with the coding team. Representative quotes are presented for the themes that identified the challenges faced by PWD.

## 3. Results

There were 3 key themes that emerged from the qualitative interviews: seamless system of care; transformative patient-to-self-agency/self-advocacy model; and collaborative research and learning environment. The first theme related to the obstacles that stakeholders identified to receiving quality healthcare services for PWD. These included lack of coordination between the healthcare sector and community well-being programs. Additionally, the panel noted that PWD face significant challenges in finding and accessing healthcare services that meet their specific needs and that they can self-manage. The panel recommended that well-being self-management programs be developed and implemented that are easily accessible from home so that PWD can be the drivers of their own health and well-being. 

### 3.1. Seamless System of Care

Stakeholders noted the need for development of a seamless system of care from acute care through rehabilitation, post-rehabilitation, and lifelong health and well-being.

(Re)Defining the Continuum (To Include Health, Well-Being, and Quality of Life)

Stakeholders articulated a need to define/re-define a seamless (no gaps between transitions) continuum of care from acute care, rehabilitation, post-rehabilitation, and home/community lifelong health promotion/well-being. A key component of this continuum is improving quality of life by supporting individuals in reaching their highest level of life functioning.

2.Coordination along the Continuum

Stakeholders shared that the healthcare system tends to break down at different points along the continuum of care. In other words, “people fall through the cracks” and have difficulty getting answers to the highly complex, big picture of their health. Stakeholders want to see this changed so care is highly coordinated along the continuum. This would include clinical improvements, collaboration among clinicians and rehabilitation professionals, education of medical professionals along the continuum, and consideration of how care translates to the home and community.

### 3.2. Transformative Patient-to-Self-Agency/Self-Advocacy Model


*Whole Person Approach*


Stakeholders focused on providing the opportunity for a person with a disability to cultivate a healthy lifestyle. This approach would include addressing physical, mental, and spiritual well-being through the integration of exercise, nutrition, and mindfulness programs and services. At the center of this is a “patient-centered, whole person approach” instead of the approach of simply treating a medical issue. The individual’s family and support network are also a key focus of the whole person approach.

2.
*Outcomes for Individual and Beyond*


Stakeholders agreed the anticipated outcome of the integration of mindfulness, exercise, and nutrition would be a self-reliant and confident person, full participation and engagement in community life and the things that are most important to the person, and returning to or reaching the highest level of life functioning. At the same time, stakeholders also focused on outcomes beyond the individual such as a healthier population that spends less on healthcare and encounters fewer readmissions for medical care. 

3.
*Technology as a Tool*


Stakeholders focused on the potential for technology to serve as a critical tool both in coordination along the continuum and serving the whole person in life-long learning endeavors. Technology could be used in a range of aspects to enhance self-agency/self-advocacy using various apps and a comprehensive telehealth platform.

4.
*Measurement and Cost*


Stakeholders highlighted the importance of developing systems to measure the cost of getting people to their highest level of life functioning. They felt this would be critical in terms of research, publications, grants, and contracting.

### 3.3. Collaborative Research and Learning Environment

The third key theme was where stakeholders emphasized that collaborative research will enrich the knowledge base around rehabilitation and post-rehabilitation health and well-being through an environment that integrates the work and contributions of students, medical professionals, researchers, patients, and others.


*Research*


Stakeholders focused on the importance of research serving as a foundation for helping to build a strong evidence base for positively affecting practice and policy on a broad scale. Particular areas of research could include developing a standard for rehabilitation outcomes, demonstrating that a model of well-being is cost-effective and beneficial to individuals, and conducting longitudinal studies in general.

2.
*Inclusive Science and Multidiscilinary Collaboration*


Stakeholders emphasized the importance of a multidisciplinary approach with public health, health professions, medicine, psychology, psychiatry, education, and engineering among others. Bringing together professionals from diverse areas of study and practice, including through **Inclusive Science**, is part of what will make these efforts unique.

3.
*Economic Development*


Stakeholders said the model of an integrated research and learning environment will also present significant economic development opportunities through its ability to attract national and international attention.

### 3.4. Individual Key Stakeholder and Partner Interviews


*Nine stakeholder and partner key informant individual interviews conducted by the consulting firm focused on the interviewee’s knowledge about the unique needs of the target population, opportunities to improve care for the target population, and challenges for implementing a healthcare-to-well-being program. Based on an iterative thematic analysis of the interviews, five overarching themes emerged. Each of these are highlighted below with exemplar quotes.*


The healthcare system is structured upon a medical model of curing illness instead of improving well-being, and this model limits care for PWD who are not “sick” in conventional medical terms.
*a*.*“We have to begin by telling people that they are not ‘sick.’ They have a disability which is quite different from the ‘sick’ model of health care that we have been accustomed to.”**b*.*“This group needs to understand the wellness versus sick model. We need to begin this process early on, because they have to learn how to take care of themselves.”*Individuals with disability need ongoing awareness and education to understand their condition and possible limitations, and caregivers or partners/family members need training to provide quality care and maintain their own well-being.
*a*.*“It is critically important for patients and their families/social networks to understand their condition and any possible limitations. The other thing is to have information on the trajectory of a condition/prognosis to better help individuals see what role the diagnosis plays in their overall lifespan.”**b*.*“Education, support, and problem solving are not unique to these populations, necessarily, but perhaps more intensive. For newly acquired disabilities, this group has a unique need in terms of understanding the resources available. Also, for those with a newly acquired disability, they have unique needs to connect with vocational rehab.”**c*.*“Caregivers’ health issues are also a burden on the health system, so they should be included in any of the offerings for health and well-being and this type of platform could be the method for including them.”*Education about well-being for PWD must also target policymakers, health service providers, and the general public.
*a*.*“Education has to be not only to the patients but also to policymakers and healthcare providers. Traditionally the doctor just says, ‘you need to lose 5 pounds,’ with no real follow-up or referral or resources, and we need to find a way to go beyond that somehow.”**b*.*“We need to get the message out that obesity is a major complicating factor for people with disabilities, and it needs to start sooner than adulthood.”*Telehealth offers an innovative way to expand medical and well-being services for PWD.
*a*.*“Access restrictions and financial limitations make telehealth very important for this population. This population would significantly benefit from being able to access medical care without having to physically get themselves to Birmingham.”**b*.*“I think the greatest opportunity will be where telehealth services intersect with patient care. This could happen by introducing telehealth when a patient is in the hospital, moving to telehealth in the home which would include patient monitoring and telehealth home visits and then could eventually move to having telehealth providers (non-MDs) supporting the patient in health and wellness (nutritionist, health coaches, etc.).”*Coordinated and integrated individually-tailored well-being plans are difficult to execute but critical for improving health outcomes for PWD.
*a*.*“Right now, everything is still pretty piecemeal. The different systems are barriers. The disabled population needs doctors and practitioners working in a team and talking about their care.”**b*.*“They need interventions that are specific to their needs. There are a lot of variations to that. In the world of people with disability (physical, visual, cognitive, etc.), there isn’t one approach that will be effective in education or intervention. Also, folks who have a disability need to be included in the development of these interventions so we can be sure interventions are as relevant as possible.”**c*.*“Time is a challenge in creating health and well-being programs. Making individualized, tailored plans for each person and having the time to really connect them with resources is a challenge.”*

## 4. Discussion

Bridging the gap between the challenges of linking the healthcare system to well-being home and community-based services requires a set of intentional, coordinated strategies to achieve better health and well-being among PWD. There is an urgent need to provide accessible and appropriate healthcare-to-well-being services and programs for people with disabilities that supports them in their well-being journey as they self-advocate and manage their health in conjunction with their healthcare providers, systems of care, and support networks. A detailed strategic plan includes recommendations, obstacles in developing close relationships with healthcare delivery organizations, and implementation procedures.

### 4.1. Recommendations

A critical component in developing the proposed framework is to expand the role of existing Inclusive Health Coalitions (IHCs) (Table 1). Traditionally, IHCs have had the role of establishing partnerships between community health organizations to create accessible health promotion programs and ensure that the inclusion of people with disability is a cornerstone in community health planning [12]. These IHCs are multi-stakeholder groups of community members working together to promote inclusion in all aspects of community life. We propose that individual IHCs expand their missions to include a specific focus on patient transitions from healthcare to well-being, helping patients progress from medically managed care in which they receive services from healthcare providers to full-fledged participation in long-term well-being. IHCs should initiate efforts to engage healthcare delivery organizations in their areas to promote inclusive well-being services with PWD as they are interacting with the healthcare system and developing their autonomy from the healthcare system. IHCs offer an existing organizational structure that can support the provision of well-being services. By providing these services and engaging healthcare delivery organizations in the creation and dissemination of these programs, IHCs can make progress in creating a unique continuum of care and help public health meet its eventual goal of ensuring the autonomy of PWD.

We believe that IHCs will need to adopt a formal structure designed to ensure a seamless transition for the individual from healthcare to well-being. This new structure should have several features including the ability to connect different organizations involved in the care of individuals with disability. These groups include:Community-based organizations focused on well-being for PWD and the general public, such as fitness and recreation centers.Healthcare providers caring for individuals with a disability.Community-based organizations that can provide support for individuals with disability facilitating their participation in well-being activities (e.g., transportation—accessible ride shares, taxis/cabs, buses; durable medical equipment providers; architects) and;Payers such as commercial health insurers, government payers (e.g., Medicare and Medicaid), state workers’ compensation programs, and private philanthropic organizations.

In addition, IHCs need to have the flexibility to adapt to differences in local environments, the capacity to offer benefits that meet the unique needs of individuals with disability, and the ability to build relationships with large, national organizations that serve as thought leaders promoting and disseminating best practices in well-being initiatives for PWD. 

To meet this new or expanded purpose, an IHC may need to engage in activities to formalize its organizational structure such as writing a clear mission statement, receiving letters of commitment or memoranda of understanding, and documenting clear expectations of partnerships. We recommend the following set of initial preliminary activities for IHCs:Identify whether existing groups share a focus on the health of PWD. If so, aim to collaborate with those that have an interest in this work.Identify diverse stakeholders to serve on the IHC. This should include PWD, caregivers, and gatekeepers identified at all levels of the SEM. PWD and their family members must be pivotal partners of the IHC and have an active voice.Establish and maintain a commitment to the IHC and its mission. Think broadly about the mission. Groups interested in health equity, participation, increasing health of all populations, disability-specific interests, and social justice initiatives may offer unique input and serve as gatekeepers that can be engaged in this work.Formally partner with organizations that advocate and serve PWD to leverage their ability to encourage their members to participate in the program. IHCs should identify a specific contact within each organization and, if that individual leaves, identify another contact to serve in their place. IHCs need to define objectives, outputs, or activities to be completed in collaboration with involved organizations to maintain connectedness to the program. For these partnerships to be successful, IHCs should establish a clear and regular communication protocol for updates, needs, and successes of the program to maintain engagement with organizations.

While IHCs have been very successful in promoting community-level change, we anticipate that the ‘loose’ organizational structure IHCs currently employ may be a barrier to developing close relationships with healthcare delivery organizations. The end goal of integrating with healthcare delivery organizations may require formal partnerships characterized by well-defined responsibilities that meet the needs of both healthcare delivery organizations and the patients that these organizations serve. These formalized relationships can help to establish permanence and accountability.

### 4.2. Detailed Strategic Plan to Implement a Disability-Inclusive Healthcare-to-Well-Being Translational Science Framework 

The Disability-Inclusive Healthcare-to-Well-Being Translational Science Framework is founded on health equity and community engagement principles geared towards building connections, collaborations, and capacities at each level of the social ecological model (SEM) using evidence-informed/evidence-based strategies for sustainable resources and networks of PWD, organizations, and institutions committed to knowledge translation and long-term well-being for PWD.

Figure 1 provides an overview and Table 2 provides a step-by-step guide for public health workers and community members to create a well-being program for PWD. It focuses on developing necessary materials by identifying health and well-being needs, the capacity of community members, and legal and governance institutions in place; intentionally co-creating with PWD and their support networks throughout the entirety as drivers of the plan. 

Further, it seeks to increase the knowledge and skill of health care providers to support the transition of patients into community-based well-being programs. A ‘boots on the ground’ committee is necessary to identify and engage with stakeholders, including community members with disability and their caregivers, family members, and service providers. The committee should ensure that individuals who have recently accessed the healthcare system for care related to a new or existing disability are intimately and intentionally involved throughout co-planning, co-executing, and co-reviewing the transitional healthcare-to-well-being program(s).

Creating awareness and consensus among stakeholders and well-being recipients as they transition into self-advocates and champions of their care are critical. This includes educating community partners and healthcare providers about programs by creating training sessions on how to best support PWD as they transition from health care to well-being. All implementing partners should be encouraged to utilize the training and cement the radical shift in public health that the programs are creating. Further, the framework recognizes that program implementation will be strengthened by using the data collected from previous phases and further engaging target populations, increasing access to programs for PWD and allies in the community. The framework seeks to advance growth of programming to make healthcare to well-being mainstream in communities. All program implementers must remain committed to the mission while remaining receptive and responsive to the changing capacities and needs of community members, ensuring a sustainable shift in public health for PWD.

There are some limitations of the proposed strategic plan. The strategic plan is based on a finite number of expert panel members and stakeholders. Accordingly, the reflections, ideas, and recommendations are limited to those who were included and may not represent all circumstances nor all disabilities. There are additional strategies that could be proposed and implemented. We believe, however, that the strategic plan offered here takes into account a vast array of disciplines, experiences, and disabilities to provide a well-crafted strategic plan that holds great promise. 

## 5. Conclusions

### Transforming Healthcare Patients into Well-Being Self-Advocates with Agency

This strategic plan is aimed at encouraging local communities to develop a healthcare-to-well-being infrastructure (i.e., coalition) to support the well-being needs of PWD and achieve parity and equity in access to public health initiatives. The plan proposes to build inclusive communities that enable PWD to self-manage or co-manage their health and well-being in their own communities. This can be accomplished by creating a structure that supports a continuum of care for PWD that would enable them to transition from medically managed healthcare to self-managed well-being using the proposed plan as part of the linkages. Through these initiatives, PWD will have greater access to, and inclusion in, community well-being programs and public health initiatives. Inclusion is a societal ideology that states that opportunities are intended to be equal for all people—which can be achieved!

## Figures and Tables

**Figure 1 ijerph-21-00018-f001:**
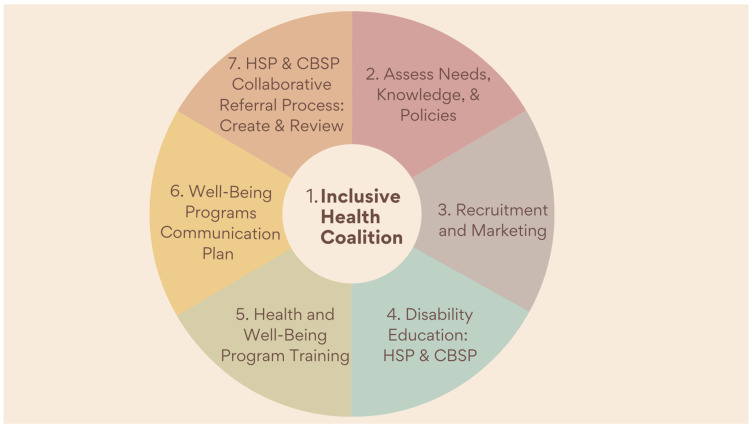
Overview of Strategic Plan to Implement a Disability-Inclusive Healthcare-to-Well-Being Translational Science Framework; Health Service Providers (HSP) and Community Based Service Providers (CBSP).

**Table 1 ijerph-21-00018-t001:** Inclusive Health Coalitions (IHCs)—Community Leaders and Organizations.

**Inclusive Health Coalitions (IHCs)—Community Leaders and Organizations**
Promote disability inclusion in programs and services related to well-being promotion.
Focus on the removal of barriers that prevent PWD achieving the same health transformation opportunities as available to anyone else in the community. Create and modify community health programs, policies, systems, and environments to include PWD.

**Table 2 ijerph-21-00018-t002:** Strategic Plan to Implement a Disability-Inclusive Healthcare-to-Well-Being Translational Science Framework: Phases, and Activities.

Phases	Activities
Create Inclusive Health Coalition (IHC)	Create IHC to: o Identify and engage PWD o Conduct focus groups and/or interviews o Create a stakeholder committee of individuals who have recently accessed the healthcare system for care related to a new or existing disability, their caregivers and family members, and local service providers Engage the individual’s support network including families and friends through events and activities
2. Assess health and well-being needs, community partners’ knowledge of PWD, and existing services and policies	Gauge health and well-being needs Examine facilitators and barriers of access to community health and well-being programs Estimate the capacity and capability to participate in well-being programs List the existing community-based health and well-being programs that PWD can join Assess existing community services that could attenuate factors that restrict access Know local, regional, state, and federal policy, regulations, and laws related to factors that facilitate or restrict the access to community-based health and well-being programs
3. Expand knowledge and facilitate successful transitions of patients into community programs and initiatives; engage stakeholders to support increasing participation of PWD	Engage in recruitment, marketing, and communication activities (available via various formats) o Identify and meet with gatekeepers of community-based service providers o Build relationships and partnerships with local health service providers and community-based service providers to access PWD o Identify communication needs and resources to reach those who are isolated o Develop a marketing and outreach plan including messaging, materials, and a list of potential channels of advertising o Develop a comprehensive recruitment plan that targets all and intentionally reaches isolated groups o Establish incentives for community health and wellness center (including fitness and recreation centers) to offer the program to PWD Facilitate access to telehealth to provide in-home programs
4. Educate community partners, specifically health service providers (HSP) and community-based service providers (CBSP) about the community well-being program	Develop communication and training plans tailored to the needs of health service providers (HSP) o Create print, video, and audio forms of education materials o Identify HSP’s concerns about their patients’ potential participation o Develop a training plan to enhance HSP’s knowledge about the program o Implement HSP communication and training plan o Deploy evaluation plan and implement the activities as needed to achieve the desired awareness and knowledge Help community-based service providers (CBSP) develop the necessary knowledge and skills to support transition of PWD into the program o Determine CBSP comfort level and interest in transition process of PWD from a health care setting to the well-being program. o Identify CBSP concerns about participation of PWD in the program and what knowledge or training would increase their comfort o Increase CBSP disability knowledge and education, universal design principles, and local, regional, and national accessibility policies and address specific concerns. o Support HSP and CBSP in implementing the transition process of PWD to the program o Deploy an evaluation plan to assess buy-in, concerns, and to identify gaps in the training plan for HSP and CBSP o Revise and implement activities as needed
5. Offer health promotion/well-being programming and training for target population and stakeholders	Provide training and/or collaborate with community partners to increase well-being training opportunities specifically designed for PWD and their family members, caregivers, and related stakeholders o Identify potential community partners, including organizations’ names, leaders, contact information, and current programs o Identify the training areas to be addressed in the program. o Engage PWD to determine their needs. o Identify gaps in current training programs. o Engage existing programs to determine openness to expansion. o Draft content for each training. o Determine schedule of trainings offered (e.g., all at once, sequentially, and/or in rotation).
6. Execute comprehensive communication plan to raise community well-being program awareness	Work closely with hospitals and rehabilitation facilities to raise awareness about the program o Identify local hospital and rehabilitation facilities that can develop a program or refer patients to an existing one. o Share examples of transitional healthcare-to-well-being programs with local hospitals and rehabilitation facilities. If none exist in the local community, encourage use of the program. o Facilitate public awareness and social marketing with the help of healthcare facilities (e.g., meet with leadership, use flyers, tap into social media, capitalize on marketing strategies that are important to the specific target group). Use mass media and online marketing tools to increase enrollment in well-being program o Engage community members for outreach. o Use strategies from existing programs to promote enrollment in the well-being program Ensure comprehensive media coverage for local and regional news and events associated with the well-being program o Identify potential local and regional news agencies and provide an overview of the well-being program to build interest. o Develop a timeline for implementation and identify which phases in well-being program implementation will leverage the most benefit from media coverage. o Reach out to local and regional news agencies for coverage of the most important components of the well-being program to boost enrollment
7. Assist HSP and CBSP to Formalize Process for Patient Referral to community well-being program	Develop and reinforce strong relationships between HSP and CBSP Support HSP and CBSP in enacting a formal procedure to transition patients from rehabilitation to post-rehabilitation well-being programs. Identify potential barriers and provide solutions and strategies to overcome those barriers Document all challenges encountered during initial implementation and solutions found

## Data Availability

Data is contained within the article.

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
