# Peer review of "A Disability-Inclusive Healthcare-to-Well-Being Translational Science Framework"

_ijerph, 2023, doi:10.3390/ijerph21010018_

Round 1

Reviewer 1 Report

Comments and Suggestions for Authors

Dear Authors,

Thank you for your important and indeed necessary work. May I please raise the following suggestions? Thank you.

a) Please explain how the stakeholders were selected. If they are stakeholders, please share their important characteristics in more detail that make them eligible to be stakeholders.

b) Please explain how the stakeholders were approached and enrolled into the research, these details will add more dimension to your paper.

c) Please explain how the stakeholders' identities were kept confidential, or was confidentiality of identity waived by the stakeholders?

d) Please explain on what basis the stakeholders were divided into two committees? Was it experience, being a PWD or was it random? This is a necessity to add more dimension to the paper.

e) Please explain who the consulting firm is? Is the consulting firm comprised of the authors? Please explain did the authors approach this research and the stakeholders as academic authors or as a consulting firm? Please clarify. If the consulting firm was engaged by the authors could you please provide more details on how this was done and how the partnership between the two parties was managed? Thank you. 

The strengths of the paper include:

a) The strategic plan and framework are data driven. This gives the suggested framework an empirical basis for validity.

b) Well founded in an academic argument that needs to be addressed. 

c) Table 2 is particularly interesting in terms of application and utility for PWD and other relevant stakeholders. 

Reviewer 2 Report

Comments and Suggestions for Authors

A very important paper which might hopefully trigger further research in the area of people with disabilities as populations are increasingly greying and citizens must learn to live autonomously despite individual challenges arising along the way.

There were some minor places which needed correction; please see attached file for suggestions.

Reviewer 3 Report

Comments and Suggestions for Authors

In response to the urgency of better identifying and systematically addressing health inequalities, this study attempted to create a strategic plan by considering the opinions of an expert panel consisting of 14 stakeholders to develop a strategic plan addressing this issue. As a result, providers and local communities are provided with a framework that will enable them to manage their health in partnership with healthcare providers to begin addressing accessible and appropriate healthcare to well-being services and programs for people with disabilities. Although this study deals with a sensitive and important issue, some deficiencies need to be eliminated for this study to reach the desired level. Some of those:

Point 1. Information about the methods and data used in the summary of the study is not provided.

Point 2. The introduction part of the study should be enriched. This study should be compared with other studies. The need and importance of this study should be mentioned.

Point 3. Brief information about the sections of the study can be given at the end of the introduction of the study.

Point 4. Qualitative coding is mentioned in the method part of the study. However, there is no detailed information about this coding.

Point 4.1. How does this coding system work?

Point 4.2. Is there a code hierarchy?

Point 4.3. How was the code set developed?

Point 4.4. What is thematic analysis and how was it done?

Point 5. Statistical results of the data obtained during the interview should be included.

Point 6. The results and discussion sections of the study are well addressed.

The authors of the study should especially provide detailed information about the method of the study.

Comments on the Quality of English Language

Minor editing of English language required

Round 2

Reviewer 3 Report

Comments and Suggestions for Authors

Accept in present form